# Improving Adherence to Myofunctional Therapy in the Treatment of Sleep-Disordered Breathing

**DOI:** 10.3390/jcm10245772

**Published:** 2021-12-09

**Authors:** Carlos O’Connor-Reina, Jose María Ignacio Garcia, Laura Rodriguez Alcala, Elisa Rodríguez Ruiz, María Teresa Garcia Iriarte, Juan Carlos Casado Morente, Peter Baptista, Guillermo Plaza

**Affiliations:** 1Otorhinolaryngology Department, Hospital Quironsalud Marbella, 29603 Marbella, Spain; carlos.oconnor@quironsalud.es (C.O.-R.); laurarod3108@gmail.com (L.R.A.); jccasadom@hotmail.com (J.C.C.M.); 2Otorhinolaryngology Department, Hospital Quironsalud Campo de Gibraltar, 11379 Palmones, Spain; 3Pulmonology Department, Hospital Quironsalud Marbella, 29603 Marbella, Spain; josemariaignacio@gmail.com (J.M.I.G.); Elyrodriguez203@hotmail.com (E.R.R.); 4Otorhinolaryngology Department, Hospital Virgen de Valme, 41014 Sevilla, Spain; garciairiartemt@hotmail.com; 5Otorhinolaryngology Department, Clínica Universitaria de Navarra, 31008 Pamplona, Spain; peterbaptista@gmail.com; 6Otorhinolaryngology Department, Hospital Universitario de Fuenlabrada, Universidad Rey Juan Carlos, 28942 Madrid, Spain; 7Otorhinolaryngology Department, Hospital Sanitas la Zarzuela, 28942 Madrid, Spain

**Keywords:** obstructive sleep apnoea, myofunctional therapy, adherence, apnea hypopnea index, Iowa oral performance instrument

## Abstract

Myofunctional therapy (MT) is used to treat sleep-disordered breathing. However, MT has low adherence—only ~10% in most studies. We describe our experiences with MT delivered through a mobile health app named Airway Gym^®^, which is used by patients who have rejected continuous positive airway pressure and other therapies. We compared ear, nose, and throat examination findings, Friedman stage, tongue-tie presence, tongue strength measured using the Iowa oral performance instrument (IOPI), and full polysomnography before and after the 3 months of therapy. Participants were taught how to perform the exercises using the app at the start. Telemedicine allowed physicians to record adherence to and accuracy of the exercise performance. Fifty-four patients were enrolled; 35 (64.8%) were adherent and performed exercises for 15 min/day on five days/week. We found significant changes (*p* < 0.05) in the apnoea–hypopnoea index (AHI; 32.97 ± 1.8 to 21.9 ± 14.5 events/h); IOPI score (44.4 ± 11.08 to 49.66 ± 10.2); and minimum O_2_ saturation (80.91% ± 6.1% to 85.09% ± 5.3%). IOPI scores correlated significantly with AHI after the therapy (Pearson *r* = 0.4; *p* = 0.01). The 19 patients who did not adhere to the protocol showed no changes. MT based on telemedicine had good adherence, and its effect on AHI correlated with IOPI and improvement in tongue-tie.

## 1. Introduction

Obstructive sleep apnoea–hypopnoea syndrome (OSAHS) is a chronic sleep-related condition, or sleep-disordered breathing (SDB), that is becoming increasingly widespread; it represents a significant health cost [1,2]. According to Eckert [3], the physiopathology of SDB involves a weak muscular response in many patients. Recent reports show that people with OSAHS frequently present with impaired sensorimotor deficits in the upper airway muscles [4]. These deficits are associated with apraxia, hypotonia, and changes in muscle fibres, which lead to early fatigue of the upper airway muscles [4]. These deficits can lead to impaired proprioceptive acuity in the upper airway muscles. An effective rehabilitation for improving this pathology may be proprioceptive training associated with visual or acoustic feedback [5].

Myofunctional therapy (MT) is a promising treatment for SDB [6,7,8,9]. MT is based on daily exercises aimed at strengthening the oropharyngeal muscles and facilitating the opening of the airway, which should help to improve adherence to continuous positive airway pressure (CPAP) [10,11,12,13]. As mentioned above, because OSAHS originates from the suboptimal function of the dilator muscles of the airway, MT is designed, theoretically, to treat the mechanism underlying this disease. Several papers have reported its effectiveness, including randomized controlled trials, meta-analyses, and reviews [6,7,8,9,10,11,12,13,14,15,16,17,18,19,20,21].

However, to follow MT, the patient is instructed to perform these exercises regularly for 20–40 min daily for at least three months. In most cases, patients complete the exercises independently at home without substantial feedback and without providing precise information about their performance of the exercises to the therapist. Thus, the main drawback of MT is the low patient adherence, which has been reported to be as low as 10% in some investigations [18,19,20,21].

Most existing mobile health (mHealth) applications (apps) for OSAHS centered on the diagnosis of snoring or OSAHS. However, only a few apps are designed to promote adherence to treatment with CPAP. This is important because 30–40% of patients discontinue CPAP after the first year of treatment [22,23,24]. Smartphone technology may be valuable for treating people with OSAHS and for promoting patient empowerment and self-management of CPAP [25,26] and MT [27,28,29].

We describe our experience with MT based on an mHealth app called Airway Gym^®^ (www.airwaygym.app, accessed on 4 December 2021) in patients with OSAHS. We discuss the factors that may help increase adherence, and efficacy with the use of this app.

## 2. Materials and Methods

Patients with moderate and severe OSAHS who rejected CPAP and other therapeutic options were selected for the study from the Sleep Units from Hospital Quironsalud Marbella and Hospital Campo de Gibraltar (Spain). The exclusion criteria included not having a smartphone or having problems with the temporomandibular joint. Our local ethics committee approved the study, and all participants gave written informed consent.

The protocol included complete ear, nose, and throat (ENT) examination, including the Friedman tongue position scale [30], tongue-tie presence using the modified Hazelbaker scale [31], tongue measurements using the Iowa oral performance instrument I+ (IOPI) [32], and body mass index (BMI).

Patients were evaluated by standard laboratory polysomnography following the guidelines of the American Academy of Sleep Medicine. We used the next recorded variables. EEG (C3-A2, C4-A1, O1-A2, O2-A1), submentonian and anterior tibial electromyogram, electrooculogram, airflow thermistor, thoracic and abdominal belts, ECG, body position detector, oxygen saturation and snoring.

Apnoea and hypopnoea were analysed and scored according to the following criteria. Hypopnoea was defined as a 30% decrease in airflow signal amplitude lasting ≥10 s and accompanied by ≥3% O_2_ desaturation. Apnoea was defined as a ≥90% decrease in airflow signal amplitude lasting ≥10 s. The oxygen desaturation index was used to quantify O_2_ desaturation ≥ 3%. The test results were used to define mild OSAHS as an apnoea–hypopnoea index (AHI) of 5–14 events/h of sleep, moderate OSAHS as 15–29.9 events/h of sleep, and severe OSAHS as ≥30 events/h of sleep. All results were examined manually blindly by the same sleep technician.

This IOPI test was used to measure variables related to tongue function. The instrument used in this test is a pressure transducer connected to a battery-operated amplifier, signal-conditioning circuit, and digital voltmeter. A peak holding circuit displays peak pressure in kPa on a digital readout. We measured the variables up to three times and recorded the highest value, as reported previously [32].

Patients were instructed to perform the exercises under the guidance of the Airway Gym^®^ app [27,28]. A descriptive video of the therapy is provided as Appendix A. A face-to-face initial session was performed to train the patient in the use of the app and the exercises. Telemedicine allowed the physician to retrieve daily information about adherence to and accuracy of the performance of the exercises by each patient. The proposed average training time was 20 min/day on five days/week. The exercise protocol included exercises to strengthen mainly the tone of genioglossus and buccinator, which are mainly responsible for the collapse of the upper airway during sleep. Adherent patients were defined as those for whom the app registered an appropriate use >15 min/day.

After ninety sessions of using of the app, a re-evaluation was performed by repeating the in-office sleep questionnaires, ENT evaluation, lingual stereognosis, PSG, and IOPI measurements. The effectiveness of using the app for performing MT was evaluated by comparing changes in the AHI as the primary outcome measure and the IOPI scores before and after MT.

The continuous variables were expressed as mean ± standard deviation (SD), and their comparisons between two groups were performed by student’s t test. The categorical variables were recorded as case number and percentage, and their differences were estimated by chi-square test. All the data analyses were performed by SPSS 18.0 software. *P* values less than 0.05 indicated the statistical significance of the results.

## 3. Results

A total of 54 patients were enrolled. After three months, 35 patients (65%) performed exercises for 15 min/day on five days/week. The participants were 28 men and seven women with an average age of 45.9 ± 17.8 years. Friedman staging was as follows: 4 patients (11.4%) were stage I, 11 patients were stage II (31.45%), 14 patients were stage III (40%), and 6 patients were stage IV (17%) (Table 1).

In the 35 adherent patients, the AHI changed significantly from 32.97 ± 1.8/h to 21.9 ± 14.5/h (*t* = 6.360, *p* < 0.05) (Figure 1 and Figure 2). The IOPI score also changed significantly: Tongue pressure increased from 44.4 ± 11.08 to 50.66 ± 10.2 kPa (*t* = −3.8, *p* < 0.05) (Figure 2). The minimum O_2_ saturation increased from 80.91% ± 6.1% to 85.09% ± 5.3% (*p* < 0.05) (Figure 3). BMI did not change significantly and was 25.81 kg/m^2^ at the baseline and 25.1 kg/m^2^ after 3 months. IOPI scores correlated significantly with AHI scores (Pearson *r* = 0.4, *p* = 0.01). Tongue-tie scores correlated significantly with IOPI and AHI scores (partial correlation = 0.441, *p* = 0.009). BMI and Friedman stage did not correlate significantly with the AHI (Pearson *r* = 0.3, *p* = 0.08).

In the non-adherent group of 19 patients, the variables analysed did not change significantly from before to after the three months (Table 2 and Figure 4). Non adherent group do on average 5, 4 (3, 8-7, 5) sessions in the first 2 weeks of recruitment.

## 4. Discussion

MT is a promising treatment for SDB and is based on daily exercises to strengthen the oropharyngeal muscles and facilitate the opening of the airway [6,7,8,9,10,11,12,13]. However, the main drawbacks of MT are the low adherence, which has been reported to be as low as 10% [18,19,20,21], and the absence of objective feedback [9,11,12].

Our first report on MT based on the mHealth app Airway Gym (www.airwaygym.app, accessed on 4 December 2021) [27] involved a study of non-adherent patients, but without any selection criteria to improve adherence. No studies have evaluated the factors influencing adherence to the exercises included in this app, and this current study was designed to address this lack of information. In the current work, adherence to MT was high, possibly because of the ease of contact with a physician and the oral and visual feedback about the patient’s performance provided by the app during each exercise. Measuring the IOPI score in monthly visits may also be helpful for objectively evaluating patient progress and promoting adherence. We believe that feedback should be provided with the IOPI score to ensure adherence to this therapy. For example, in patients who do not improve after using the app, the IOPI score may indicate the use of an inaccurate technique when performing the exercises.

There is controversy about the selection of exercises used in MT [21]. We may consider three kinds of patients based on myofunctional disorders: one group with conventional orofacial myofunctional disorders (tongue protrusion, tongue thrusting, etc.), a second with upper airway muscle hypotony, and a third with both disorders [21]. These differences may explain some of the heterogeneity in the results reported in the literature. We believe that the exercises performed with this app are different because they focus on proprioceptive rehabilitation and increasing muscle tone. Proprioception (or kinesthesia) is the sense through which we perceive the position and movement of our body, including our sense of equilibrium and balance, senses that depend on the notion of force [33]. When upper airway muscles contacted with the screen of the smart phone promote proprioceptive training. As we have reported previously, we consider these sensory motor rehabilitation [34] exercises when used to treat SDB, and MT as conventional exercises performed to treat conventional disorders, such as swallowing disorders.

We found significant correlations between final AHI, tongue-tie presence (Hazelbaker modified score), and IOPI scores for the tongue and lip in adherent patients. We believe a normal Hazelbaker score promotes adherence to this therapy. The initial IOPI scores were lower in the adherent group than in the non-adherent group. We hypothesize that improvement occurs earlier in patients with low IOPI scores and that this improvement contributes to the adherence to this treatment. Such patients may have the hypotonic phenotype of SDB [3]. Another study by our group [35] tried to identify the hypotonic phenotype based on the scores obtained with the IOPI and the tongue digital spoon [36] combined with the score obtained with the orofacial myofunctional evaluation with scores protocol (OMES) [37]. We hypothesize that both scores can provide complementary information for OMES, which, to our knowledge, is the only test to identify myofunctional disorders in people with SDB.

We used the Hazelbaker modified score (quantitative scale) instead of the Marchesani protocol (presence or absence of tongue-tie) to provide more accurate information about the functional importance of the tongue in the performance of these exercises. Patients with difficulties in tongue movements (Figure 5) may have poorer adherence to myofunctional therapy; however, this has not been reported. We note that examination of tongue-tie is not included in any otolaryngological clinical guidelines for evaluating the upper airway in the assessment of SDB [1,38]. However, we firmly believe it should be included.

Several papers have shown that the IOPI, in addition to PSG, can be used to provide objective feedback on the effects of therapy. Suzuki et al. [13] reported a longitudinal study of 32 patients undergoing six months of MT. The AHI decreased significantly from 34.7 to 29.0 events/h (*p* = 0.03), and tongue pressure increased significantly from 35.9 to 45.6 kPa (*p* < 0.01). Seven patients (22%), including six of the 12 with moderate OSAHS (50%), successfully discontinued CPAP. In a randomized trial with patients with severe OSAHS who used the Airway Gym^®^ app, O’Connor et al. [28] found that the AHI decreased by 53.4%, from 44.7 (95% CI 33.8–55.6) to 20.88 (95% CI 14.02–27.7) events/h (*p* < 0.001). The tongue pressure increased from 39.83 (95% CI 35.32–45.2) to 59.06 (95% CI 54.74–64.00) kPa (*p* < 0.001). The AHI score correlated significantly with tongue pressure. The authors suggested that patients without nasal obstruction or restriction of tongue movement and low airway muscle tone (as diagnosed using the IOPI) are the best candidates for MT. Using a similar instrument as the IOPI, Kim et al. [29] recently reported on another mHealth app to help patients with swallowing difficulties. After eight weeks, eight patients were adherent, and their swallowing tongue pressure significantly increased from 17.5 kPa to 26.5 kPa.

Although further evidence of the efficacy of this app is needed, we consider that this therapy may help to improve adherence to other treatments [26,28] and outcomes in patients with poor adherence to conventional treatment [27,28]. However, patients must perform the exercises contained in the app correctly. We have observed that therapy can be suboptimal when the phone or head are not placed in the correct position or move during the exercise, preventing the application of optimal muscle activity.

The limitations of our study are the small number of patients and the short follow-up time. Long-term studies are needed to understand the duration of adherence and the effectiveness of MT delivered using the app, and develop guidelines for maintenance therapy after the initial use of this app.

## 5. Conclusions

In our experience, MT based on telemedicine delivered using the Airway Gym^®^ app has good adherence and efficacy. As a form of sensory motor rehabilitation, the app positively affects AHI, and this effect correlates with changes in the IOPI and modified Hazelbaker score. Low initial scores for tongue strength measures are associated with better adherence to this therapy.

## Figures and Tables

**Figure 1 jcm-10-05772-f001:**
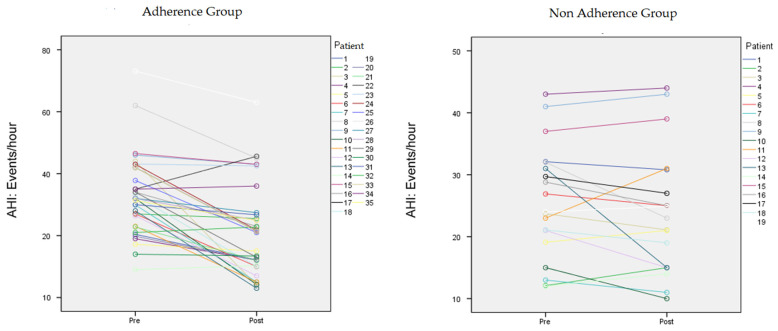
Changes in AHI in adherent and non adherent patients. AHI (events/h) changes in individual patients between the baseline and the 3-month value on the left.

**Figure 2 jcm-10-05772-f002:**
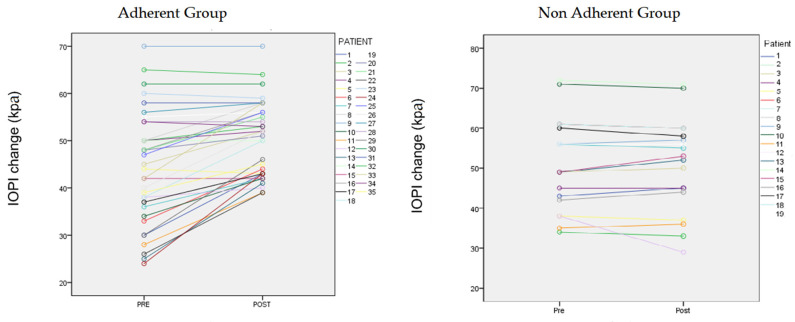
IOPI (kPa) changes in individual patients from the baseline to the 3-month value on the left.

**Figure 3 jcm-10-05772-f003:**
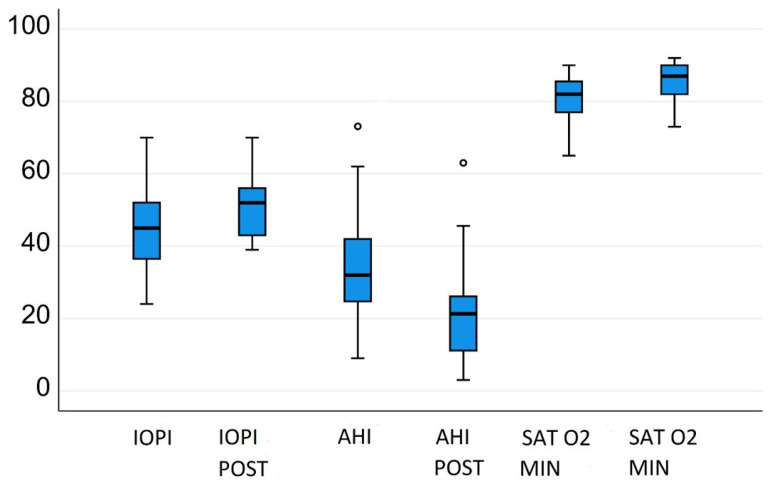
Changes in AHI, IOPI, and Sat O_2_ min in adherent patients after 3 months of exercises.

**Figure 4 jcm-10-05772-f004:**
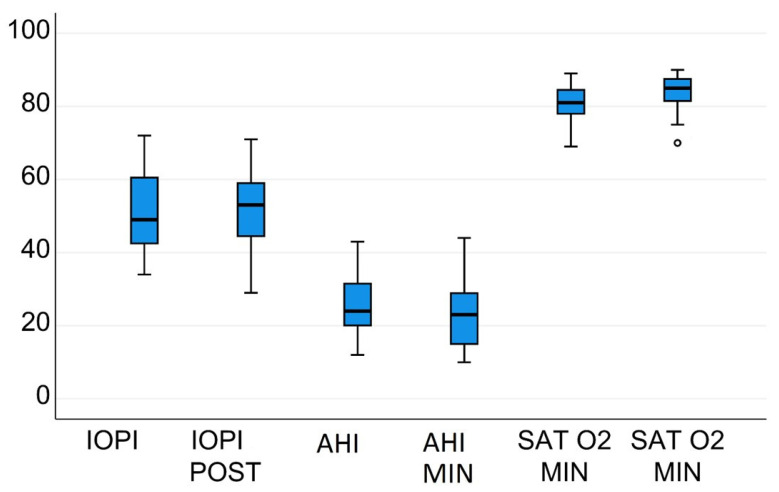
Changes in AHI, IOPI, and Sat O_2_ min in non-adherent patients after 3 months of exercises.

**Figure 5 jcm-10-05772-f005:**
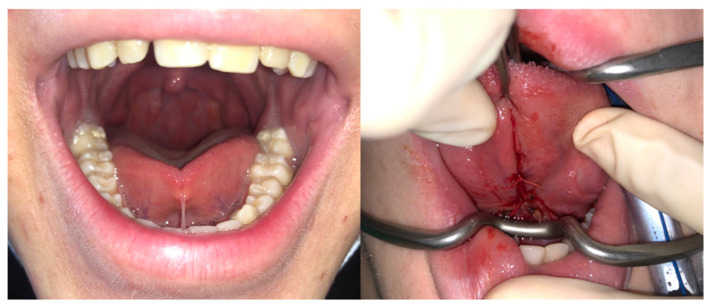
An example of severe tongue-tie at the baseline (**left**), which was improved after surgery (**right**).

**Table 1 jcm-10-05772-t001:** Baseline Data for adherent and nonadherent patients.

	Non-Adherent (*n* = 19)	Adherent (*n* = 35)	*p V*alue
**Anthropometric data**			
Age (years)	50.26 (36.2–64.224)	45.9 (28.17–63.77)	NS
Female (%)	7 (36.8%)	6 (17.14%)	NS
BMI (kg/m^2^)	25.1 ± 2.8	25.8 (22.7–29.1)	NS
Friedman stage	5 (I). 4 (II). 3 (III). 3 (IV)	4 (I). 11 (II). 14 (III). 6 (IV)	NS
Polysomnography data			
AHI/h	25.5 ± 9.2	32.97 ± 1.8/h	NS
Sat O_2_ Min	80.68 ± 5.6	80.91% ± 6.1%	NS
**Tone measures (kPa)**			
IOPI max tongue	51.3 ± 11.4	44.4 ± 11.08	0.04
27			
**Tongue-tie examination**			
Hazlebaker score	10.79 ± 2.4	13.03 ± 1.5	0.04

BMI = body mass index; AHI = apnea–hypopnea index; ODI = oxygen desaturation index; IOPI max tongue = Iowa Oral Performance Instrument maximum tongue elevation strength.

**Table 2 jcm-10-05772-t002:** Changes in variables from the baseline to the follow-up in the control and airway gym groups.

	Non-Adherent (*n* = 19)			Airway Gym Group (*n* = 35)		
	Baseline	After 3 Months	*p* Value	Baseline	After 3 Months	*p* Value
**Anthropometric data**						
BMI (kg/m^2^)	25.1 ± 2.8	24.6 ± 2.5	n.s	25.8 (22.7–29.1)	25.1.2(22.4–28)	n.s
**Polysomnographic data**						
AHI/h	25.5 ± 9.2	23.8 ± 10.1	n.s	32.97 ± 1.8/h	21.9 ± 14.5/h	0.01
Sat O _2_Min	80.68 ± 5.6	81.3 ± 5.7	n.s	80.91% ± 6.1%	85.09% ± 5.3%	0.01
**Tone measures (kPa)**						
IOPI max tongue	51.3 ± 11.4	51.1 ± 11.7	n.s	44.4 ± 11.08	50.1 ± 10.2 kPa	0.01

BMI = body mass index; AHI = apnea–hypopnea Index; Sat O_2_ min = minimal O_2_ desaturation; IOPI max tongue = Iowa Oral Performance Instrument maximum tongue elevation strength.

## Data Availability

Available on reasonable demand.

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
