# Peer review of "Improving Adherence to Myofunctional Therapy in the Treatment of Sleep-Disordered Breathing"

_jcm, 2021, doi:10.3390/jcm10245772_

Round 1

Reviewer 1 Report

The manuscipt report the results of a pre-post study aiming to investigate the efficacy of and the adherence to a telemedicine-based myofucntional therapy. The study is well written and easy to read. Myofunctional therapy has been introduced for the treatment of OSAHS in the last decade and its application is increasingly popular thanks to the non-invasive nature and the promising efficacy. The problem of adherence to the treatment is of greatest importance in OSAHS, thus, the paper address a very relevant topic. Nevertheless, some aspects should be addressed. In particular, the treatment protocol should be described in detail.

Specifically:

  • page 1 line 26: "3 months after the treatment", this sentence is misleading as it may suggest that a follow-up after 3 months from the end of treatment was performed. I would suggest to revise the sentence
  • page 1 line 35 a "not" is missing: "The 19 patients who did NOT adhere"
  • page 2 lines 47-49 the sentence needs references
  • page 2 lines 54-56: the review does not conclude that proprioceptive training is the "best" approach as no comparison was performed with other treatment approaches. Thus, the sentence should be revised
  • the IOPI procedure for tongue pressure measurement should be described
  • page 3 line 118: please check the AHI cut-offs for mild OSAHS (5-14.9, not <5)
  • the lack of a detailed description of the treatment is the main weakness of the study. The exercise protocol should be reported (in the text or appendix/supplementary material), otherwise the rationale for the efficacy of the treatment can't be appraised (e.g. what muscles are targeted, how are the pharyngeal muscles involved by the exercises).
  • Apparently, a face-by-face initial session was performed to train the patient of the use of the app and on the exercises, but it is not described in the methods
  • Results are well described and the graphs including the individual changes of AHI and IOPI are very informative.
  • Figures 3 and 4: IOPIT, what does the T stand for?
  • Table 2: the "control group" should be renamed "non-adherent group"
  • Information on the exercise amount the non-adherent group performed would be interesting.
  • Page 5 lines 253 "without any selection criteria to improve adherence": what is the meaning of this sentence?
  • Page 5 lines 258-259: it is unclear whether the authors perform a monthly re-assessment using IOPI; this has not been reported in the methods
  • page 5 lines 264: the authors states that "we considered three kinds of patients..." but this concept is novel and no reference to these 3 categories has been done previously in the paper
  • the authors define their treatement as "proprioceptive". However, again, the lack of a description of the treatment does not allow to understand how the proprioception was trained in this study
  • page 6 lines 275-276: "The initial IOPI scores were lower in the non-adherent group than in the adherent group": This is in contrast with table 1 and with the subsequent sentences
  • page 7 line 326: lip pressure measurement was not performed.

Author Response

Review 1

The manuscipt report the results of a pre-post study aiming to investigate the efficacy of and the adherence to a telemedicine-based myofucntional therapy. The study is well written and easy to read. Myofunctional therapy has been introduced for the treatment of OSAHS in the last decade and its application is increasingly popular thanks to the non-invasive nature and the promising efficacy. The problem of adherence to the treatment is of greatest importance in OSAHS, thus, the paper address a very relevant topic. Nevertheless, some aspects should be addressed. In particular, the treatment protocol should be described in detail.

Specifically:

  • page 1 line 26: "3 months after the treatment", this sentence is misleading as it may suggest that a follow-up after 3 months from the end f treatment was performed. I would suggest to revise the sentence
  • before and after the 3 months of therapy.
  • page 1 line 35 a "not" is missing: "The 19 patients who did NOT adhere"
  • The 19 patients who did not adhere
  • page 2 lines 47-49 the sentence needs references
  • Recent reports show that people with OSAHS frequently present with impaired sensorimotor deficits in the upper airway muscles [4].
  • page 2 lines 54-56: the review does not conclude that proprioceptive training is the "best" approach as no comparison was performed with other treatment approaches. Thus, the sentence should be revised
  • An effective rehabilitation for improving this pathology may be proprioceptive training
  • the IOPI procedure for tongue pressure measurement should be described
  • This IOPI test was used to measure variables related to tongue function. The instrument used in this test is a pressure transducer connected to a battery-operated amplifier, signal-conditioning circuit, and digital voltmeter. A peak holding circuit displays peak pressure in kPa on a digital readout. We measured the variables up to three times and recorded the highest value, as reported previously [32].
  • page 3 line 118: please check the AHI cut-offs for mild OSAHS (5-14.9, not <5)
  • to define mild OSAHS as an apnoea–hypopnoea index (AHI) of 5–14 events/h
  • the lack of a detailed description of the treatment is the main weakness of the study. The exercise protocol should be reported (in the text or appendix/supplementary material), otherwise the rationale for the efficacy of the treatment can't be appraised (e.g. what muscles are targeted, how are the pharyngeal muscles involved by the exercises).
  • A descriptive video of the therapy is provided as supplementary material.
  • Apparently, a face-by-face initial session was performed to train the patient of the use of the app and on the exercises, but it is not described in the methods
  • A face-to-face initial session was performed to train the patient in the use of the app and the exercises.
  • Results are well described and the graphs including the individual changes of AHI and IOPI are very informative.
  • Figures 3 and 4: IOPIT, what does the T stand for?
  • Corrected
  • Table 2: the "control group" should be renamed "non-adherent group"
  • Done
  • Information on the exercise amount the non-adherent group performed would be interesting.
  • We added Non adherent group do in average 5,4 (3,8-7,5) sessions in the first 2 weeks of
  • Page 5 lines 253 "without any selection criteria to improve adherence": what is the meaning of this sentence?
  • In our initial study, we did not select patients or encourage the use of the App.
  • Page 5 lines 258-259: it is unclear whether the authors perform a monthly re-assessment using IOPI; this has not been reported in the methods
  • In this study, the IOPI was used after three months of therapy.
  • Measuring the IOPI score in monthly visits may also be helpful for objectively evaluating patient progress and promoting adherence
  • page 5 lines 264: the authors states that "we considered three kinds of patients..." but this concept is novel and no reference to these 3 categories has been done previously in the paper
  • We may consider three kinds of patients based on myofunctional disorders: one group with conventional….(21), Carrasco LLatas, ; O’Connor-Reina, C.; Calvo-Henríquez, C. The Role of Myofunctional Therapy in Treating Sleep-Disordered Breathing: A State-of-the-Art Review. Int. J. Environ. Res. Public Health202118, 7291. https://doi.org/10.3390/ijerph18147291
  • the authors define their treatement as "proprioceptive". However, again, the lack of a description of the treatment does not allow to understand how the proprioception was trained in this study
    • Proprioception (or kinesthesia) is the sense though which we perceive the position and movement of our body, including our sense of equilibrium and balance, senses that depend on the notion of force [34]. When upper airway muscles contacted with the screen of the smart phone promote proprioceptive training.
  • page 6 lines 275-276: "The initial IOPI scores were lower in the non-adherent group than in the adherent group": This is in contrast with table 1 and with the subsequent sentences
  • The initial IOPI scores were lower in the adherent group than in the non-adherent group.
  • page 7 line 326: lip pressure measurement was not performed.
  • Low initial scores for tongue strength measures are associated with better adherence to this therapy.

Reviewer 2 Report

I think this is a very valuable research that describes the importance of myofunctional therapy(MT) for upper airway muscles and the method to improve the adherence to MT in obstructive sleep apnea.

However, there are some points to be improved.

1)  Regarding the research design, the authors compared the two groups who were adherent to the MT with Airway Gym(AG) or not, but the definition of "adherent" to the MT is not clear from the paper.  I would recommend describing it clearly.

2) In the Abstract, the 3rd line from the bottom, "The 19 patients who did adhere..." must be "The 19 patients who did not adhere...".

3) Figure1. What is the notation "iah post IAH"? In the figure legend, "...the 3month value on the right." should be "...the 3month value on the left."  I feel like individual slope, for example, would be better (same for figure2). I was also wondering if no one in the adherent group showed increase of AHI?

4) Figure3. What is the notation "o19"?  IAH should be described as AIH.

5) Figure4. What is the notation "o9"?  IAH should be described as AIH.

6) Table2, the name of each group should be "Non adherent" and "Adherent ", respectively.

7) In the discussion part, line 275, the authors describe that the initial IOPI scores were lower in the non-adherent group, but it was opposite, wasn't it?

Author Response

Review 2

Open Review

English language and style

( ) Extensive editing of English language and style required
( ) Moderate English changes required
( ) English language and style are fine/minor spell check required
(x) I don't feel qualified to judge about the English language and style

Yes

Can be improved

Must be improved

Not applicable

Does the introduction provide sufficient background and include all relevant references?

(x)

( )

( )

( )

Is the research design appropriate?

( )

(x)

( )

( )

Are the methods adequately described?

( )

(x)

( )

( )

Are the results clearly presented?

( )

(x)

( )

( )

Are the conclusions supported by the results?

( )

(x)

( )

( )

Comments and Suggestions for Authors

I think this is a very valuable research that describes the importance of myofunctional therapy(MT) for upper airway muscles and the method to improve the adherence to MT in obstructive sleep apnea.

However, there are some points to be improved.

1) Regarding the research design, the authors compared the two groups who were adherent to the MT with Airway Gym(AG) or not, but the definition of "adherent" to the MT is not clear from the paper. I would recommend describing it clearly.

Adherent patients were defined as those for whom the App registered an appropriate use >15 min/day.

2) In the Abstract, the 3rd line from the bottom, "The 19 patients who did adhere..." must be "The 19 patients who did not adhere...".

The 19 patients who did not adhere to the protocol showed no changes.

3) Figure1. What is the notation "iah post IAH"? In the figure legend, "...the 3month value on the right." should be "...the 3month value on the left." I feel like individual slope, for example, would be better (same for figure2). I was also wondering if no one in the adherent group showed increase of AHI? We have corrected it. We have modified Figure 1 and Figure 2 to include changes in adherent and non adherents groups

4) Figure3. What is the notation "o19"? IAH should be described as AIH.

Corrected. We have supressed o19

5) Figure4. What is the notation "o9"? IAH should be described as AIH.

Corrected. We have supressed o9

6) Table2, the name of each group should be "Non adherent" and "Adherent ", respectively.

Done

7) In the discussion part, line 275, the authors describe that the initial IOPI scores were lower in the non-adherent group, but it was opposite, wasn't it?

Corrected
